energy/mechanical engineering

three-roller tube expander, finite-element numerical simulation, expansion unit, taper angle, bearing limit

**Author for correspondence:**
Gang Bi
e-mail: big@xsyu.edu.cn

# Structure parameter optimization and bearing limit analysis of the expansion unit of three-roller tube expander

Gang Bi[1,2,3], Zhan Qu[1,2], Zhenquan Wang[3], Liangbin Dou[1] and Mengmeng Li[1]

[1]College of Petroleum Engineering, Xi'an Shiyou University, Xi'an, China
[2]State Key Laboratory of Petroleum Resources and Prospecting, China University of Petroleum, Beijing, China
[3]Shaanxi Key Laboratory of Well Stability and Fluid & Rock Mechanics in Oil and Gas Reservoirs, Xi'an Shiyou University, Xi'an, China

GB, 0000-0003-3393-956X

The critical technical issues for the structure design of three-roller tube expander were first studied and analysed in this paper. Then, the major design parameters of the expansion unit structure and the bearing limit of 12¼″ three-roller tube expander were optimized and investigated by finite-element numerical simulation method. Results from study show that the required expansion force increases when the taper angle of the roller outer surface gets larger, taking the axial expansion force as the quantitative indicators. It is suggested that the roller tape angle of the expansion unit should be in the range of 9–12° considering the proper length of the roller and the non-self-locking tube expansion process. The required expansion force of the bellows first decreases and then increases when the gauge length of the expansion unit becomes longer. The optimal value of the gauge length is 50 mm considering the proper length of the roller. And according to the numerical simulation results, the designed three-roller tube expander meets the strength requirements. The results of this study are of great significance to the expend bellows drilling technology.

## 1. Introduction

The oil and gas exploration and development in China is gradually developing towards offshore deep drilling. Thus, it is necessary to drill deep wells to meet the requirements of deep

water exploration and development. Due to the increase of casing used in deep wells, the drilling cost is increased. Moreover, the size of borehole is reduced by the extensive use of casing, which has a negative effect on the later drilling and completion operations [1,2]. Integrated drilling and completion is an effective method to solve this problem, and the development of expander technology allows this method to be implemented. The technology of equal-diameter isolation of complex well segments by expandable bellows is to roll the circular tube, of which the inner diameter is larger than the borehole diameter, into a bellows with diameter smaller than the borehole diameter according to the metal plastic deformation. After running the bellows into the pre-packer section, the bellows is expanded to the circular tube state by hydraulic and mechanical force to seal the complex well section [3–8]. Hydraulic expansion expands the main body of bellows to certain roundness. However, the upper and lower joint of bellows still maintains the original size, which is necessary to use mechanical expansion to inflate it to the required diameter. Expansion tool is the key of the mechanical expansion process for expandable bellows. Its structure directly influences the mechanical expansion process of bellows. If the tool structure design is unreasonable, the bellows may be damaged in the process of expansion and the expansion operation cannot be completed. Furthermore, it may result in some severe drilling accidents. Therefore, it is essential to optimize the design of mechanical expansion tools.

The mechanical expansion tools commonly used at present include three-roller tube expander and spherical expander [9–11]. The role of three-roller tube expander is to trim diameter of the bellows string assembly to the size of the corresponding borehole bit. That is, to expand the bellows upper and lower joint to the size of the corresponding borehole bit. Meanwhile, the bellows has certain ovality after hydraulic expansion, so the bellows is trimmed to the corresponding borehole size of the drill bit by three-roller expander. After the inner diameter of bellows string is expanded to the size of the corresponding borehole bit by the three-roller tube expander, the function of spherical expander is to further enlarge the inner diameter of the bellows, which makes the bellows diameter 3 mm larger than that of the drill bit, so that the drill bit can pass through the bellows well section successfully. It is indicated that three-roller tube expander is the key to the mechanical expansion process of the expandable bellows, and the rationality of its structure plays an important role in the mechanical expansion effect. Therefore, it is necessary to optimize the structural parameters of the three-roller tube expander. In this paper, according to the function requirement of expansion tools in the mechanical expansion process of bellows, the key issues of the structure design of three-roller tube expander are studied. It is indicated that the expansion unit structure is the main factor influencing the behaviour of three-roller tube expander. The major design parameters and key parts of the expansion unit structure of 12¼″ three-roller tube expander were optimized by theoretical analysis and finite-element numerical simulation method. The bearing limit of three-roller tube expander was also studied by numerical simulation method. And the three-roller tube expander can also be used in the field of coil-wound heat exchangers [12].

## 2. Key to the design of three-roller tube expander

The function of the three-roller tube expander is to trim the diameter of the bellow string assembly to the size of the corresponding borehole bit. That is, both the bellows upper and lower joint and the bellows need to be trimmed to the size of the corresponding borehole bit. If the roller tube expander completes the above two tasks, it must have two functions, including tube expansion function and rounding function.

There must be a certain taper between the tube body contact surface and the tube expansion unit of the expander for the tube expansion function of the three-roller tube expander. The thin end diameter should be less than the size of the joint inner diameter of the bellows. The big end diameter should be equal to the corresponding borehole bit size. At the same time, in order to maintain the deformation of the expanded tube body, a proper gauge length section is required, as shown in figure 1. In order to facilitate the process of running in or pulling out the tube expander and mud circulation, the circumference of the maximum outer diameter of the tube expansion unit cannot be a continuous entirety, and the convex ribbed discontinuous circumference design is required, as shown in figure 2. The rounding function is to expand the elliptical bellows body to the extent that the short axis of the ellipse is equal to the diameter of the corresponding borehole bit. The structure of the expansion unit is closely related to the rounding effect. Therefore, the configuration structure design of the tube expansion unit of the three-roller tube expander is an essential part of the tube expander design.

The maximum outer diameter of the tube expansion unit is a discontinuous circumference. Expanding or trimming the bellows body needs to rotate downward the body. Thus, the tube expander is subjected to the combined action of bit pressure and torque in the process of tube expansion. The magnitude of the bit

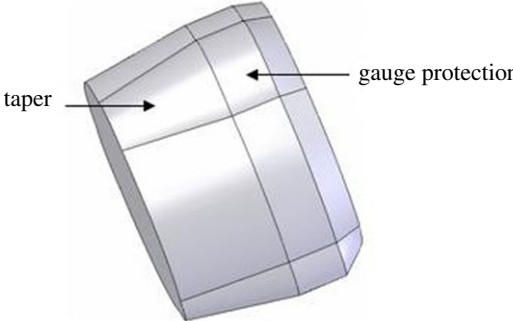

**Figure 1.** The tube expansion unit schematic diagram of the three-roller tube expander.

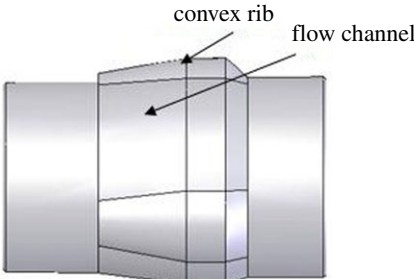

**Figure 2.** The maximum outer diameter circumference schematic diagram of the tube expansion unit.

pressure and torque depends on the strength of the upper and lower joints of the bellows, the structure of the tube expander and the joint contact unit and the form of tube expansion [13,14]. The joint strength is an objective constant value; therefore, the magnitude of bit pressure and torque mainly depends on the design of the structure and motion state of the tube expansion unit. In order to prevent the distortion or rotation of the bellows string assembly caused by high bit pressure and torque in the process of expansion, the bellows body must be effectively expanded and the bit pressure or torque required for tube expansion must be minimal for the structure and motion state design of the tube expansion unit. Based on the above analysis, the key issue of the three-roller tube expander design is the structure design of the tube expansion unit and the form of the tube expansion.

# 3. Tube expansion unit of three-roller tube expander

## 3.1. Tube expansion form optimization of expansion unit of three-roller tube expander

There are two main tube expansion forms for the tube expansion unit of three-roller tube expander:

(i) Squeezing type: the expansion unit is integrated with the housing of three-roller tube expander. There is no relative motion between them. The tube expands when the outer surface of the expansion unit squeezes the tube body, as shown in figure 3.
(ii) Rolling type: the expansion unit and three-roller tube expander housing are separated structures, and there exists relative motion between them. The outer surface of the expansion unit rotates under the action of the expansion force while it revolves around the shell of the tube expander. The tube expands when the surface of the expansion unit rolls to the tube body, as shown in figure 4. According to different bearing system of the expansion unit, the structure of the expansion unit can be divided into two types, including sliding type and rolling type.

The friction form of the contact part is different, due to the different expansion form of the expansion unit. The friction on the friction surface of squeezing type expansion form is sliding friction, while on the friction surface of rolling type is rolling friction. The rolling friction force is less than sliding friction force, and thus, the required torque for rolling type tube expansion is less than that of the squeezing type

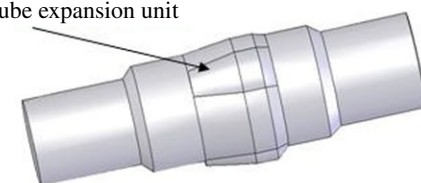

**Figure 3.** Squeezing type three-roller tube expander.

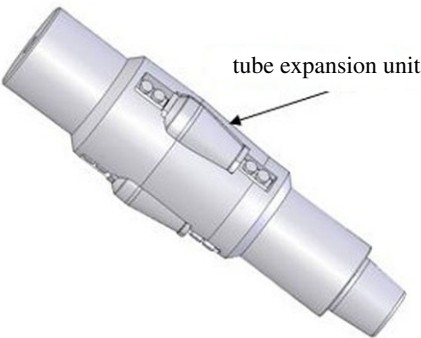

**Figure 4.** Rolling type three-roller tube expander.

expansion. In this paper, the expansion form for the expansion unit of three-roller tube expander is rolling type, which is referred as a roller tube expander.

## 3.2. Expansion unit structure design of three-roller tube expander

The expansion unit of three-roller tube expander is mainly made of two parts, including roller and roller bearing system, as shown in figure 5.

The roller structure parameters include: big end diameter, small end diameter, taper and gauge length. The big and small end diameters are determined by joint size, borehole diameter and roller number. The taper of roller outer surface and gauge length, which have a direct effect on the pressure required for tube expansion, are the main parameters that need to be optimized. In this paper, the process of bellows expansion by three-roller tube expander is simulated by finite-element numerical simulation method. The expansion force is taken as the quantitative indicator to optimize the gauge length and the outer surface taper of the roller.

The roller bearing system is composed of rolling bearing group and thrust bearing. The lifespans of the bearing system become focus of the study due to the complex bearing load of the bearing system and poor lubrication condition in the expansion process. The bearing capacity and structural strength are considered in the design of the rolling bearing group. The stepped hole is designed for the roller inner cavity in order to improve the roller structural strength. Three groups of needle roller bearings are used to improve the bearing capacity of the bearing system. The size of needle roller bearing is designed according to the optimal bearing capacity.

The thrust bearing is composed of the upper and the lower thrust parts. The force on the lower thrust is small during the working process, and thus, the design focuses on the upper thrust bearing [15,16]. The upper thrust bearing bears the reaction force generated by squeezing the tube wall in the working process. Meanwhile, the structure of thrust bearing is an open structure due to the limitation of the space structure and the friction surface of the bearing directly connected with the borehole annulus. The mud is used as lubricating and cooling fluid; the bearing surface is badly abrasive worn by the solid phase particles in the mud. Therefore, it is necessary to improve the bearing capacity of the thrust bearing and the abrasive resistance.

The sliding friction form is adopted to improve the bearing capacity of the thrust bearing. The friction material on friction surface needs to be optimized to improve the abrasive resistance capacity of the friction surface. According to the performance indicators of commonly used super-hard materials, it can be indicated that diamond-based materials are the hardest materials and its friction coefficient is low. Therefore, it is an ideal friction surface material for abrasive resistance. Polycrystalline diamond is selected as the friction surface material so as to enhance the friction surface abrasive resistance of the thrust bearing of the expansion unit.

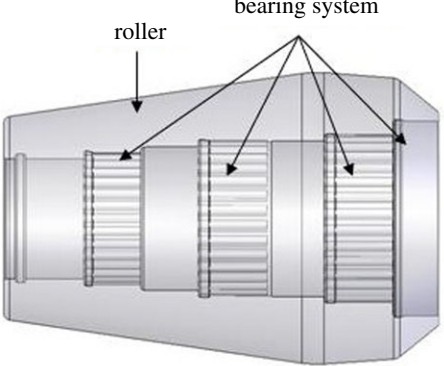

**Figure 5.** The expansion unit of three-roller tube expander.

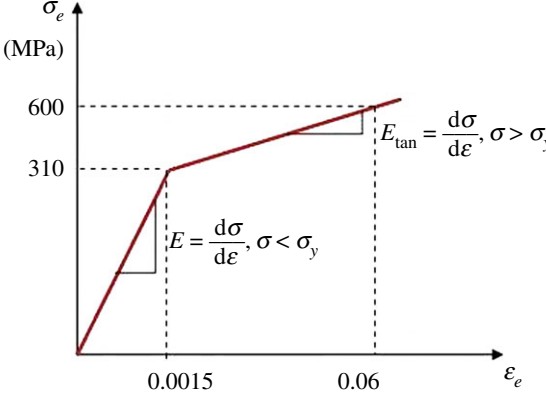

**Figure 6.** Bilinear isotropic kinematic material model.

# 4. Structure parameter optimization of the expansion unit of three-roller tube expander

## 4.1. Constitutive model

The stress–strain curves of bellows material were obtained by tensile tests [17,18]. The relationship between real stress and strain and theoretical stress and strain is shown in the following formulae:

$$\sigma_{ts} = \frac{fl}{A_o A_c} = \sigma_e(1 + \varepsilon_e) \tag{4.1}$$

and

$$\varepsilon_{ts} = 1n\left(\frac{l}{l_o}\right) = 1n(1 + \varepsilon_e), \tag{4.2}$$

where $\sigma_{ts}$ is real stress of the bellows, MPa; $\varepsilon_{ts}$ is real strain of the bellows.

As shown in figure 6, the material was hardened along the line after the material yielding during finite-element analysis according the selected material model. Therefore, the tangent modulus and material physical parameters should be defined, such as density, Poisson ratio and yielding point.

## 4.2. Geometric model of three-roller tube expander and bellows

The size of three-roller tube expander is 12¼″ in this paper. The structure of the tube expander is complex. If using the actual structure of the expander to simulate the bellows expansion process, it needs great amount of calculation and it is too complex to complete the calculation. Moreover, the main purpose of simulating the mechanical expansion process of bellows by finite-element numerical simulation method is to carry out strength design and size design for the main parts and main

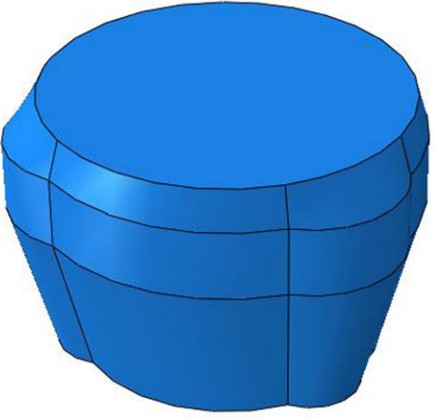

**Figure 7.** Schematic diagram of the structure of three-roller tube expander.

**Table 1.** Structure parameters of bellows.

| type | inner diameter (mm) | outer diameter (mm) | wall thickness (mm) | tube length (mm) |
|------|---------------------|---------------------|---------------------|------------------|
| 12¼″ | 281 | 311 | 15 | 500 |

**Table 2.** Structure parameters of tube expansion unit.

| type | outer diameter (mm) | gauge length (mm) | taper (°) |
|------|---------------------|-------------------|-----------|
| 12¼″ | 311 | 0, 20, 50, 80, 100, 150 | 3, 8, 12, 15, 20, 30 |

parameters of the tube expander. Therefore, it is necessary to simplify the expander structure. That is, to simplify the structure of the joint tube expander into a roller tube expansion unit to simulate the expansion process, as shown in figure 7. Hydraulic expansion can inflate the main body of the bellows to certain roundness, but the upper and lower joints of the bellows still maintain the original size. And after hydraulic expansion, the bellows surface will still be uneven and elliptic.

The main function of the roller tube expander is to expand the joints of bellows. Therefore, the bellows is simplified into a round tube. According to the structure, parameters of bellows and tube expander are listed in tables 1 and 2. The finite-element numerical model for the interaction between bellows and tube expander is shown in figure 8.

The type of material unit is SOLID185 type unit, which is a three-dimensional solid unit model, according to calculation accuracy requirements. The interaction model between the tube expander and bellows is divided by VSWEEP, which can divide model into regular hexahedral unit. The hexahedral unit has two more calculation sections than the conventional tetrahedral unit, which can increase the calculation accuracy and reduce the computation time.

## 4.3. Definition and control of contact surface

There exists frictional wear in the process of bellows deformation under the action of tube expander. Therefore, the contact behaviour between the tube expander and bellows must be considered.

Contact behaviour is divided into two types: rigid body–soft body contact and semi-soft body–soft body contact. In this paper, the tuber expander is stiffer than the bellows. Therefore, the contact behaviour is rigid body–soft body contact.

ABAQUS has three contact forms, including point–point contact, point–surface contact and face–face contact. Each contact form has its own contact unit. The contact unit of bellows and tube expander is a face–face contact unit. The tube expander surface is regarded as the 'target' surface and the bellows surface is regarded as the 'contact' surface. The nature of the contact surface is defined and the friction coefficient of the contact surface is set to 0.3.

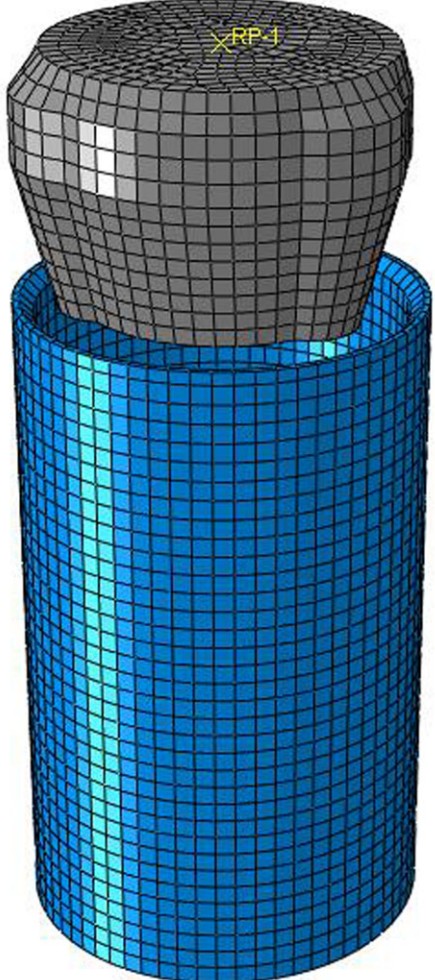

**Figure 8.** Interaction model between three-roller tube expander and bellows.

## 4.4. Loading and boundary constraint setting

Constraint of bellows: it is assumed that there is no constraint outside of the bellows, which means that the contact between bellows and well wall is not considered, and there is no displacement on the plane perpendicular to the wellbore axis for the bellows.

Constraint of expansion unit: the expansion unit moves downward in the operation process. Assuming the speed of the downward movement is constant, the movement of expansion unit can be described by uniform movement along the wellbore axis. According to the actual drilling parameters, the rotate speed is 40 r.p.m. and drilling speed is 2 m h$^{-1}$. The moving downward velocity, $v$, along the axial direction is 0.56 mm s$^{-1}$ through calculation. The moving distance of expansion unit is assumed as 200 mm. The friction coefficient between the expansion unit surface and the tube expander inner wall is 0.3. The displacement loading is shown in figure 9.

## 4.5. Result analysis of numerical simulation

Keep the gauge length of 50 mm constant, and change the taper of roller expansion unit to simulate the expansion process. The taper value are set to 3°, 8°, 12°, 15°, 20° and 30°, respectively. The von Mises stress cloud diagrams of the expanded bellows are shown in figure 10.

The von Mises stress directly influences the bearing capacity of bellows. From the figure, the maximum value of the von Mises stress with different tapers of the expansion unit is equal after bellows expansion. The value is 465 MPa. The taper has little effect on the maximum value of the von Mises stress. However, the variation trend of the von Mises stress is different when the taper angle is different. The von Mises stress is basically the same in gauge length section with different taper angle

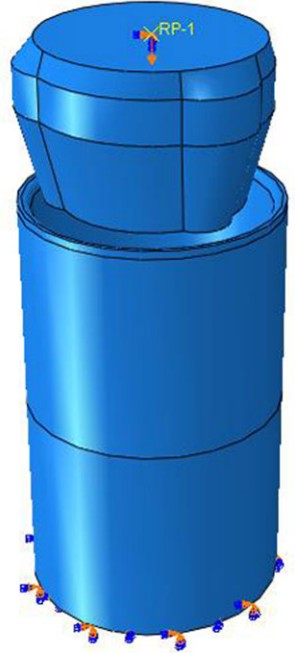

**Figure 9.** Schematic diagram of model displacement loading.

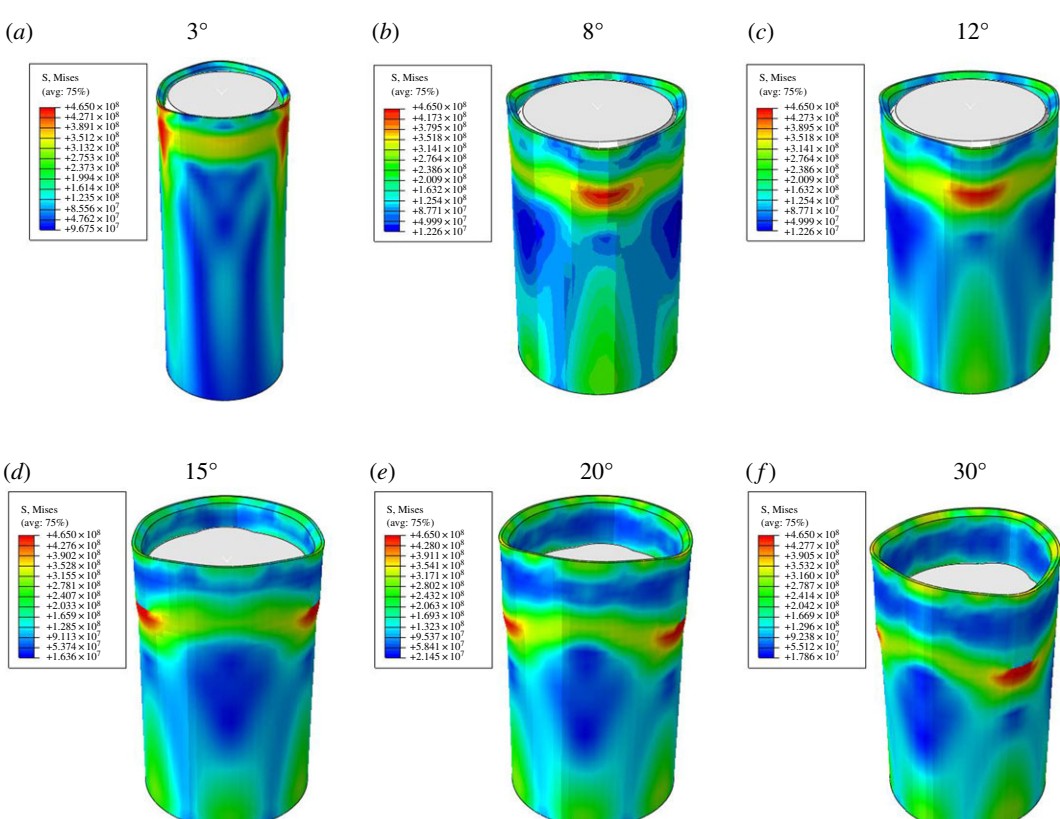

**Figure 10.** von Mises stress cloud diagram of the expanded bellows.

of the expansion unit. In taper section, the von Mises stress increases and the change trend gets greater with the increase of taper angle.

The axial expansion force gradually becomes larger as the movement displacement of expansion unit increases in the expansion process. The expansion force suddenly changes after the expansion unit contacts with bellows. After that, the axial expansion force changes rapidly with the increase of the outer diameter of expansion unit. When outer diameter reaches the maximum value, the variation

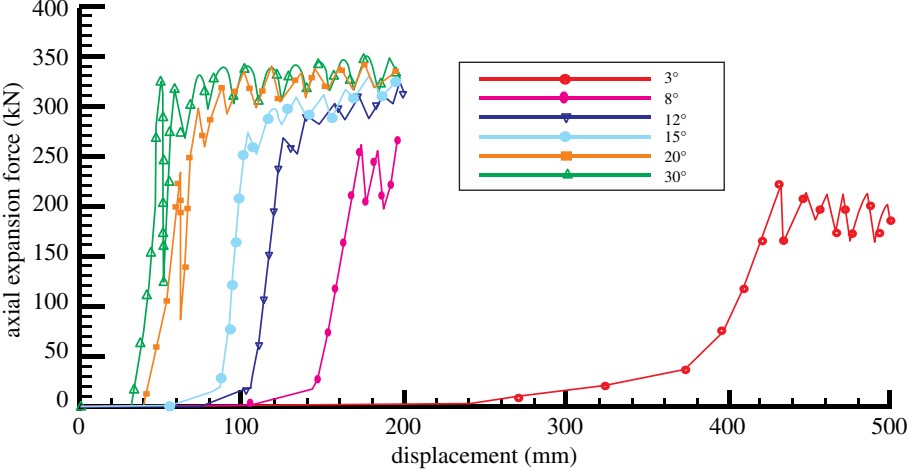

**Figure 11.** The variation of axial expansion force with displacement under different taper angle conditions.

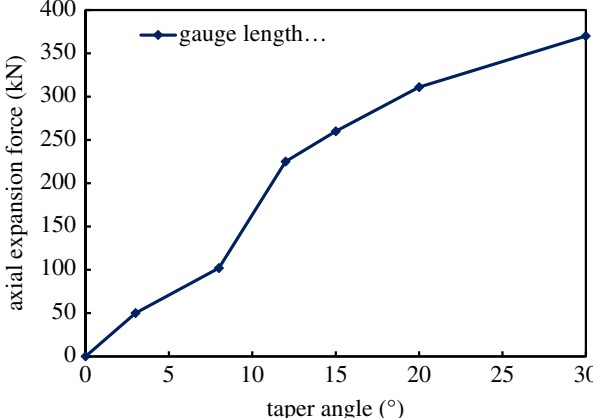

**Figure 12.** The change of axial expansion force with taper angle.

trend becomes gentle. After reaching the gauge length section, the axial expansion force gradually maintains steady state. When the taper angle of expansion unit is different, the relationship between displacement and axial expansion force is shown in figure 11. The figure shows that the expansion force of the expansion unit with different taper increases with the increase of displacement in bellows expansion process and the growth trend is basically the same. The smaller the taper angle is, the smaller the maximum expansion force needed. When the taper angle is large, the axial expansion force will have big fluctuation and become unstable in the process of bellows contacting with the expansion unit.

The force required for the tube expander to expand the bellows can be obtained from the numerical simulation results. The specific pressure value can be calculated by solving the reaction force through finite-element method. Results from calculation show that the tube expansion force increases when the taper angle of the roller outer surface becomes larger, as is shown in figure 12. Therefore, a smaller taper angle should be used in order to decrease expansion pressure. However, if the taper is too small, it will form self-locking in the tube expansion process, and the roller length will be increased. Taking all the above factors into account, the range of the taper angle is 9–12° in actual design. In this paper, the taper angle is set as 12°.

Keep the taper angle of expansion unit of 12° constant, and calculate the axial expansion force distribution with the change of displacement when the gauge lengths are 0, 20, 50, 80, 100 and 150 mm, as shown in figure 13. From the figure, the curves of axial expansion force for bellows expansion shows a V shape with different gauge length, as shown in figure 14. That is, the required expansion force first drops and then rises when the gauge length of expansion unit increases. When the gauge length is 0 mm, the required maximum expansion force reaches up to 460 kN. The reason is that the tube exerts a resilience force after expansion when the expansion unit has no gauge length

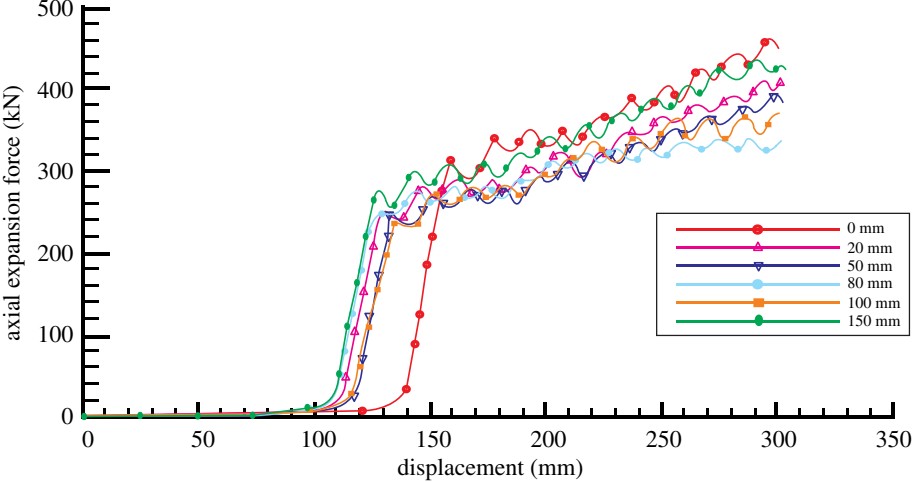

**Figure 13.** The variation of axial expansion force with displacement under different gauge length conditions.

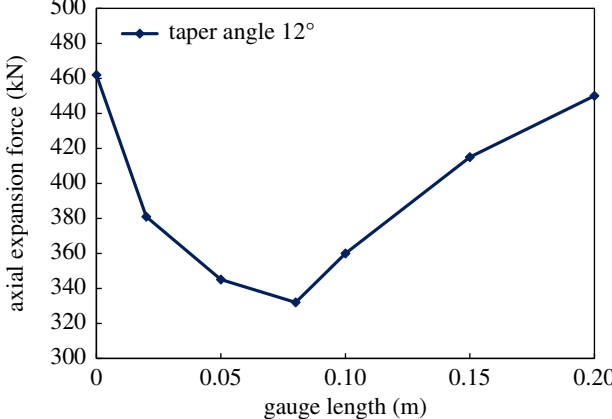

**Figure 14.** The change of axial expansion force with gauge length.

section. The loading on the expansion cone has to overcome the resilience force to make the expansion force reach maximum value. When the gauge length is 80 mm, the required expansion force is the smallest. When the gauge length is less than 80 mm, the resilience force of the bellows is the main influencing factor and the effect of the friction force is relatively small. Therefore, the expansion force decreases with the increase of the gauge length when the value of sizing length is from 0 to 80 mm. When the gauge length is more than 80 mm, the friction force becomes the control factor, leading the required expansion force to go up again. Taking the roller length factor into account, the gauge length is designed as 50 mm in the paper.

# 5. Strength check

The three-roller tube expander stands the combined action of bit pressure and torque during the working process and forms four basic stress states, including lifting, lowering, rotating lifting and rotating lowering. According to the different stress state of the roller expander, the strength of the roller expander structure is checked by finite-element numerical calculation. There are two purposes for strength check. One is to choose the needle roller bearing to further optimize the tube expander structure. The other is to verify the reliability of the expander components. The housing strength of the three-roller tube expander is mainly checked in this paper.

The strength check of the housing is to get the rated operating load of the housing under different stress states. The four stress states are simulated for strength check by finite-element numerical simulation method. The finite-element analysis models for different operation processes are shown in figures 15–18.

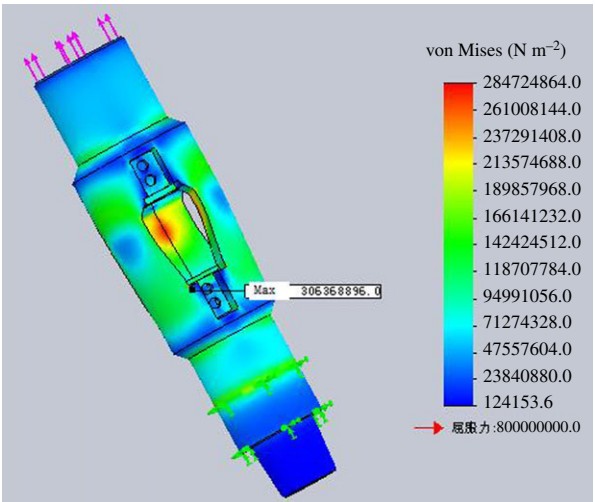

**Figure 15.** Diagram of finite-element analysis when lifting the housing.

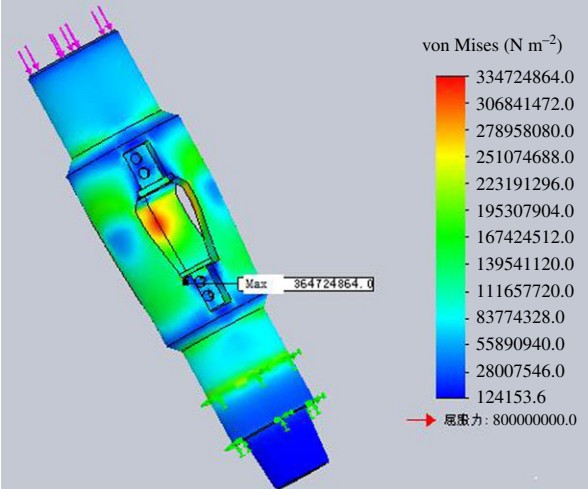

**Figure 16.** Diagram of finite-element analysis when lowering the housing.

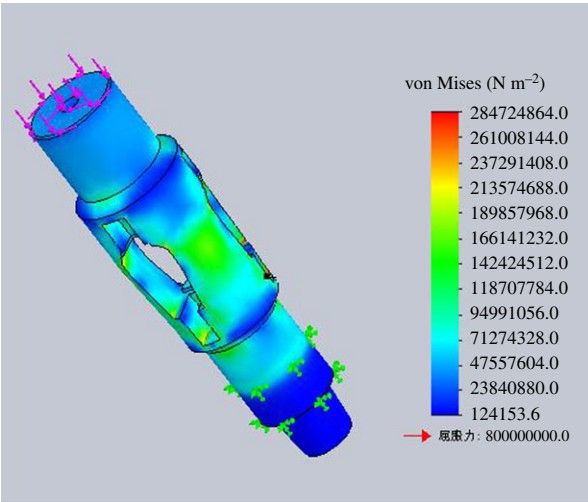

**Figure 17.** Diagram of finite-element analysis when rotating lowering the housing.

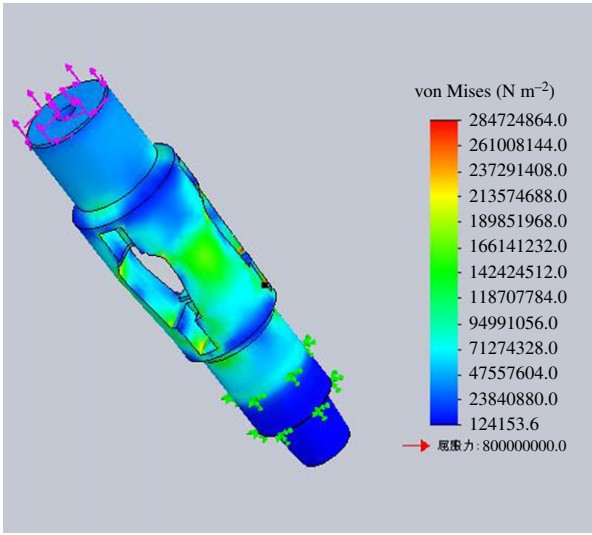

**Figure 18.** Diagram of finite-element analysis when rotating lifting the housing.

**Table 3.** The relationship between rated bit pressure and rated torque in rotating lowering process.

| bit pressure (kN) | torque (kN m) | safety factor ($n$) |
| --- | --- | --- |
| 20 | 55.0 | 2 |
| 30 | 54.2 | 2 |
| 40 | 53.4 | 2 |
| 50 | 52.5 | 2 |
| 100 | 51.0 | 2 |

According to the finite-element numerical simulation results, when the minimum safety coefficient of the inner wall of the tube expansion unit is 2, it can be regarded as the bearing limit of the housing in the process of lifting and lowering. The inner wall of the expansion unit groove is the weakest part of the housing through numerical analysis.

Assuming the minimum safety factor is 2, the maximum allowable lifting force for the housing is 2000 kN when lifting the housing. And the maximum allowable bit pressure for the housing is 2500 kN when lowering the housing. The maximum allowable torque for the housing is 55 kN m when rotating the housing.

The rated working torque with different bit pressure, of which the value is within the range of rated working bit pressure, in rotating lifting process is shown in table 3. Through finite-element numerical simulation for the expansion process of the tube expansion unit, the axial expansion force is 322.25 kN, which is smaller than the maximum lifting force and maximum bit pressure that the housing can bear. The bit pressure or lifting force is 322.5 kN in the process of rotating lifting or rotating lowering of the housing. When the safety factor is 2, the torque that the housing can bear is 30 N m, which meets the strength requirement of the housing.

## 6. Conclusion

(i) The configuration structure of the tube expansion unit of three-roller tube expander is designed considering the influence of tube expansion function and the lifting and lowering process. It is proposed that the design of the tube expansion unit structure and the motion state must be in the premise of effectively expanding the bellows body and regarding the required minimum bit pressure or torque for the tube expander as the target. Therefore, the rolling type tube expander body is used for the roller structure of the three-roller expander.

(ii) The finite-element ABAQUS software was used to simulate the mechanical expansion process of the bellows. The expansion unit structure parameters of the roller expander are optimized taking the axial expansion force as the quantitative indicator. According to the comprehensive factors of non-self-locking in tube expansion process and the proper roller length, the roller taper angle of the expander unit should be between 9° and 12°. And the roller gauge length was optimized as 50 mm.

(iii) Through the analysis of the bearing and lubrication conditions of the bearing system, the roller bearing system is designed considering the bearing capacity and structural strength of the bearing. According to the overloading and severe abrasive wear conditions of the bearing thrust, it is proposed that the emphasis of thrust bearing design is the bearing capacity and abrasive resistance of the bearing friction surface. In order to improve the abrasive resistance of the friction surface material, the polycrystalline diamond is selected as the friction surface material of the thrust bearing.

(iv) Based on the optimization of the roller expander structural parameters, the strength of the 12¼″ three-roller tube expander housing was checked. According to the numerical simulation results, the designed three-roller tube expander meets the strength requirements.

Permission to carry out fieldwork. We do not need any permissions prior to conducting our research.

Data accessibility. Data are available within the Dryad Digital Repository: https://doi.org/10.5061/dryad.j3tx95x8b [19].

Authors' contributions. G.B. conceived of the study, designed the study, coordinated the study and helped draft the manuscript. Z.Q. carried out the statistical analyses. Z.W. participated in the design of the study and drafted the manuscript. L.D. carried out the numerical simulation research and participated in data analysis. M.L. carried out the numerical simulation research and participated in data analysis. All authors gave final approval for publication.

Competing interests. We have no competing interests.

Funding. This study was supported by Foundation of State Key Laboratory of Petroleum Resources and Prospecting; China University of Petroleum, Grant/Award no. PRP/open-1807 (G.B., Z.W. and L.D.); National Natural Science Foundation of China, Grant/Award nos. 51704237, 51674200 and 51604224 (G.B., Z.Q. and M.L.) and Natural Science Foundation of Shaanxi Province, Grant/Award nos. 2019JQ-403 and 2018JM5017 (G.B. and L.D.)

Acknowledgements. The authors gratefully acknowledge the Youth Innovation Team of Shaanxi Universities.

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
