## [Reviewer comments · Royal Society Open Science]

Review History

RSOS-191630.R0 (Original submission)

Review form: Reviewer 1

Is the manuscript scientifically sound in its present form?

Yes

Are the interpretations and conclusions justified by the results?

Yes

Is the language acceptable?

Yes

Do you have any ethical concerns with this paper?

No

Have you any concerns about statistical analyses in this paper?

No

Recommendation?

Accept with minor revision (please list in comments)

Comments to the Author(s)

Thank you for submitting, this is an interesting manuscript and it should be published. Here a few minor comments that may help further improving this work:

Abstract:

Please clearly state what you trying to investigate here - try to make this more concise. What are you numerically simulating - what were the results you obtained.

Introduction:

- another large market for tube expanders are heat exchangers - could your research be relevant for this market as well?

General Comments:

- What are the underlying formulars used for your calculations? I think you should mention at least 3 and compare the results to other results from the literature.

- Haneklaus et al. (<https://www.sciencedirect.com/science/article/abs/pii/S0924013616300644>) see file attached, list a number of different methods to estimate the applied pressure.

Please try to compare to results obtained with this data

You could cite this article and the other articles listed there. At the moment you have a large number of Chinese references, which may not represent all the research that was done in this area

Review form: Reviewer 2**Is the manuscript scientifically sound in its present form?**

Yes

Are the interpretations and conclusions justified by the results?

Yes

Is the language acceptable?

Yes

Do you have any ethical concerns with this paper?

No

Have you any concerns about statistical analyses in this paper?

No

Recommendation?

Accept with minor revision (please list in comments)

Comments to the Author(s)

This paper studies and analyzes the critical technical issues for the structure design of three-roller tube expander. It is indicated that the expansion unit structure is the main influencing factor for the performance of three-roller tube expander. The major design parameters and key parts of the expansion unit structure of 12 1/4" three-roller tube expander was optimized by theoretical analysis and finite element numerical simulation method. The research results are of great significance and reference value for the development of expandable bellows drilling.

Overall, the article is well organized and its presentation is excellent. However, some minor issues still need to be improved:

1. Pay attention to the format specification of the pictures in the article, especially pictures 10 and 12.
2. Read through the full text carefully, and try to modify the words, sentences or paragraphs that are ambiguous or confusing, and strive to describe refinement, smoothness and accuracy.
3. The conclusion part of the manuscript should be rewritten to emphasize the importance of investigated subject.
4. Please keep the reference style uniform and add some recent references.
5. There are some Chinese words in Figure 15.

Decision letter (RSOS-191630.R0)

14-Feb-2020

Dear Professor Bi,

The editors assigned to your paper ("Structure Parameter Optimization and Bearing Limit Analysis of the Expansion Unit of Three-roller Tube Expander") have now received comments from reviewers. We would like you to revise your paper in accordance with the referee and Associate Editor suggestions which can be found below (not including confidential reports to the Editor). Please note this decision does not guarantee eventual acceptance.

Please submit a copy of your revised paper before 08-Mar-2020. Please note that the revision deadline will expire at 00.00am on this date. If we do not hear from you within this time then it will be assumed that the paper has been withdrawn. In exceptional circumstances, extensions may be possible if agreed with the Editorial Office in advance. We do not allow multiple rounds of revision so we urge you to make every effort to fully address all of the comments at this stage. If deemed necessary by the Editors, your manuscript will be sent back to one or more of the original reviewers for assessment. If the original reviewers are not available, we may invite new reviewers.

- Data accessibility

<http://datadryad.org/submit?journalID=RSOS&manu=RSOS-191630>

- Competing interests

- Authors' contributions

- Acknowledgements

- Funding statement

Best regards,

Lianne Parkhouse

Editorial Coordinator

on behalf of the Associate Editor, and Professor R. Kerry Rowe (Subject Editor)

Associate Editor's comments:

Two reviewers have provided commentary on your manuscript, which you should pay close attention to and provide responses to in both the revised (and clearly marked up) manuscript and your point by point response. It has been noted that your manuscript may benefit from support from professional language editors, examples of which may be found at <https://royalsociety.org/journals/authors/benefits/language-editing/>. P:llkea

Reviewers' Comments to Author:

Reviewer: 1

Comments to the Author(s)

Thank you for submitting, this is an interesting manuscript and it should be published. Here a few minor comments that may help further improving this work:

Abstract:

Please clearly state what you trying to investigate here - try to make this more concise. What are you numerically simulating - what were the results you obtained.

Introduction:

- another large market for tube expanders are heat exchangers - could your research be relevant for this market as well?

General Comments:

- What are the underlying formulars used for your calculations? I think you should mention at least 3 and compare the results to other results from the literature.
 - Haneklaus et al. (<https://www.sciencedirect.com/science/article/abs/pii/S0924013616300644>) see file attached, list a number of different methods to estimate the applied pressure.
 Please try to compare to results obtained with this data

You could cite this article and the other articles listed there. At the moment you have a large number of Chinese references, which may not represent all the research that was done in this area

Reviewer: 2

Comments to the Author(s)

This paper studies and analyzes the critical technical issues for the structure design of three-roller tube expander. It is indicated that the expansion unit structure is the main influencing factor for the performance of three-roller tube expander. The major design parameters and key parts of the expansion unit structure of 12 1/4" three-roller tube expander was optimized by theoretical analysis and finite element numerical simulation method. The research results are of great significance and reference value for the development of expandable bellows drilling.

Overall, the article is well organized and its presentation is excellent. However, some minor issues still need to be improved:

1. Pay attention to the format specification of the pictures in the article, especially pictures 10 and 12.
2. Read through the full text carefully, and try to modify the words, sentences or paragraphs that are ambiguous or confusing, and strive to describe refinement, smoothness and accuracy.
3. The conclusion part of the manuscript should be rewritten to emphasize the importance of investigated subject.
4. Please keep the reference style uniform and add some recent references.

5. There are some Chinese words in Figure 15.

Author's Response to Decision Letter for (RSOS-191630.R0)

See Appendix A.

RSOS-191630.R1 (Revision)

Review form: Reviewer 1

Is the manuscript scientifically sound in its present form?

Yes

Are the interpretations and conclusions justified by the results?

Yes

Is the language acceptable?

Yes

Do you have any ethical concerns with this paper?

No

Have you any concerns about statistical analyses in this paper?

No

Recommendation?

Accept as is

Comments to the Author(s)

Well done

Review form: Reviewer 2

Is the manuscript scientifically sound in its present form?

Yes

Are the interpretations and conclusions justified by the results?

Yes

Is the language acceptable?

Yes

Do you have any ethical concerns with this paper?

No

Have you any concerns about statistical analyses in this paper?

No

Recommendation?

Accept as is

Comments to the Author(s)

This article is well modified and can be published as it is now.

Decision letter (RSOS-191630.R1)

Dear Professor Bi,

It is a pleasure to accept your manuscript entitled "Structure Parameter Optimization and Bearing Limit Analysis of the Expansion Unit of Three-roller Tube Expander" in its current form for publication in Royal Society Open Science. The comments of the reviewer(s) who reviewed your manuscript are included at the foot of this letter.

on behalf of the Associate Editor and Professor R. Kerry Rowe (Subject Editor)
openscience@royalsociety.org

Associate Editor Comments to Author:

Thanks for addressing the reviewers' concerns.

Reviewer comments to Author:

Reviewer: 1
Comments to the Author(s)

Well done

Reviewer: 2

Comments to the Author(s)

This article is well modified and can be published as it is now.

Appendix A

Dear Editor,

On behalf of my co-authors, we thank you very much for giving us an opportunity to revise our manuscript, we appreciate editor and reviewers very much for their positive and constructive comments and suggestions on our manuscript entitled “Structure Parameter Optimization and Bearing Limit Analysis of the Expansion Unit of Three-roller Tube Expander” (ID: RSOS-191630).

We have studied reviewer’s comments carefully and have made revisions which are highlighted in red in the revised manuscript. We have tried our best to revise our manuscript according to the comments. Attached please find the revised version and point by point responses to the reviewers’ comments which we would like to submit for your kind consideration.

We hope that the revised version of the manuscript is now acceptable for publication in your journal. Looking forward to hearing from you.

Thank you and best regards.

Yours sincerely,

Gang Bi

Point by point responses to the reviewers’ comments:

Reviewer: 1

Comments to the Author

(1) Abstract: Please clearly state what you trying to investigate here - try to make this more concise. What are you numerically simulating - what were the results you obtained.

Response: According to reviewer's opinion, the abstract was revised and refined in detail, which were marked in red in the 1st page of the revised manuscript.

(2) Introduction: another large market for tube expanders are heat exchangers - could your research be relevant for this market as well?

Response: In this paper, the expandable bellows technology is mainly applied in the field of oil drilling engineering. As for the heat exchangers, it may have common properties with the expandable bellows. However, I think they should be two different tools.

(3) **What are the underlying formulars used for your calculations? I think you should mention at least 3 and compare the results to other results from the literature.**

Response: According to reviewer's opinion, we have added related formulas for our calculations, which were marked in red in the 7th page of the revised manuscript.

(4) Haneklaus et al. see file attached, list a number of different methods to estimate the applied pressure. Please try to compare to results obtained with this data. You could cite this article and the other articles listed there. At the moment you have a large number of Chinese references, which may not represent all the research that was done in this area.

Response: In this paper, the expandable bellows technology is mainly applied in the field of oil drilling engineering. In Haneklaus's article, the three roller tube expander is used in the field of coil-wound heat exchangers. We think that they are two completely different areas. And Haneklaus's article has been cited in the 3rd page of the introduction.

Reviewer: 2

Comments to the Author

(1) Pay attention to the format specification of the pictures in the article, especially pictures 10 and 12.

Response: According to reviewer's opinion, we have modified the format specification of the pictures, including 10 and 12, which was marked in red in the 12th and 14th page of the revised manuscript.

(2) Read through the full text carefully, and try to modify the words, sentences or paragraphs that are ambiguous or confusing, and strive to describe refinement, smoothness and accuracy.

Response: According to reviewer's opinion, we have modified and refined the full text in detail and strive to describe refinement, smoothness and accuracy.

(3) The conclusion part of the manuscript should be rewritten to emphasize the importance of investigated subject.

Response: According to reviewer's opinion, we have rewritten the conclusion part of the manuscript, which was marked in red in the 17th and 18th page of the revised manuscript.

(4) Please keep the reference style uniform and add some recent references.

Response: According to reviewer's opinion, we have carefully checked and modified the reference style and added some recent references, which was marked in red in the 20th page of the revised manuscript.

(5) There are some Chinese words in Figure 15.

Response: According to reviewer's opinion, we have modified the picture 15, which was marked in red in the 16th page of the revised manuscript.